# Co-Occurrence of Hepatitis A Infection and Chronic Liver Disease

**DOI:** 10.3390/ijms21176384

**Published:** 2020-09-02

**Authors:** Tatsuo Kanda, Reina Sasaki, Ryota Masuzaki, Hiroshi Takahashi, Taku Mizutani, Naoki Matsumoto, Kazushige Nirei, Mitsuhiko Moriyama

**Affiliations:** Division of Gastroenterology and Hepatology, Department of Medicine, Nihon University School of Medicine, 30-1 Oyaguchi-kamicho, Itabashi-ku, Tokyo 173-8610, Japan; sasaki.reina@nihon-u.ac.jp (R.S.); masuzaki.ryota@nihon-u.ac.jp (R.M.); hiroshi.t.215@gmail.com (H.T.); mattakunotaku1981@yahoo.co.jp (T.M.); matsumoto.naoki@nihon-u.ac.jp (N.M.); nirei.kazushige@nihon-u.ac.jp (K.N.); mizutani.taku@nihon-u.ac.jp (M.M.)

**Keywords:** HBV, HCV, HIV, acute liver failure, nonalcoholic fatty liver diseases, NASH, GRP78

## Abstract

Hepatitis A virus (HAV) infection occasionally leads to a critical condition in patients with or without chronic liver diseases. Acute-on-chronic liver disease includes acute-on-chronic liver failure (ACLF) and non-ACLF. In this review, we searched the literature concerning the association between HAV infection and chronic liver diseases in PubMed. Chronic liver diseases, such as metabolic associated fatty liver disease and alcoholic liver disease, coinfection with other viruses, and host genetic factors may be associated with severe hepatitis A. It is important to understand these conditions and mechanisms. There may be no etiological correlation between liver failure and HAV infection, but there is an association between the level of chronic liver damage and the severity of acute-on-chronic liver disease. While the application of an HAV vaccination is important for preventing HAV infection, the development of antivirals against HAV may be important for preventing the development of ACLF with HAV infection as an acute insult. The latter is all the more urgent given that the lives of patients with HAV infection and a chronic liver disease of another etiology may be at immediate risk.

## 1. Introduction

Liver failure is a common disease with high mortality, and its incidence is increasing with the use of alcohol and the prevalence of obesity and diabetes [1,2,3]. It has also been reported that the prognosis of acute hepatitis or acute liver injury was affected by the preexistence of chronic liver diseases and cirrhosis [1,2], extrahepatic diseases, such as metabolic, malignant, and psychiatric diseases [4], and host factors, such as older age and obesity [3,5,6], although the etiology of acute insults is one of the most important risk factors for the development of severe liver diseases [1,7].

Hepatitis A virus (HAV) infection is still one of the major causes of acute hepatitis worldwide. HAV infection occasionally causes acute liver failure [4,8]. It has been reported that a superinfection of HAV in patients with a chronic hepatitis C virus (HCV) infection is associated with fulminant hepatitis [9], although much research denies this association [5,10]. HAV infection rarely causes acute liver failure in patients without underlying chronic liver diseases [9].

There are excellent, safe, and effective HAV vaccines to prevent HAV infection. However, HAV vaccination costs a lot. As no universal vaccination program against HAV infection exists in certain countries, such as Japan, it may be important to develop potential drugs against HAV infection [11].

In this review, we searched the recent literature concerning the association between HAV infection and chronic liver diseases, including metabolic associated fatty liver disease (MAFLD), in PubMed. We also discussed the mechanism of severe acute hepatitis A.

## 2. Acute-On-Chronic Liver Failure with HAV Infection as an Acute Insult

Acute-on-chronic liver diseases include acute-on-chronic liver failure (ACLF) and non-ACLF [12]. ACLF, which presents acutely with multiple organ failure and is precipitated by an acute insult, has high short-term mortality [2,13]. In general, the prognosis of ACLF is worse than that of acute liver failure. ACLF is a distinct concept, where acute hepatic decompensation occurs in patients with chronic liver disease or cirrhosis in encountering an acute insult, leading to high short-term mortality [2]. In Asian countries, hepatitis viruses are important factors of acute insults, unlike in European countries and the United States [2], and HAV is one of the acute insults of ACLF [1,12,14,15,16,17].

HAV superinfection was found to be the most common etiology (42%) of acute deterioration in children with ACLF in India [15]. ACLF in adults was found to be due to HEV, HAV, or both in 61%, 27%, and 6% of cases [1], respectively, although HAV infections occur in childhood, and HAV infection as an acute insult in adult ACLF is relatively uncommon in India [17]. Agrawal et al. reported an adult patient with ACLF and HAV as an acute insult who had an underlying cirrhotic liver due to nonalcoholic steatohepatitis (NASH) [17]. Among the children and adults with ACLF, acute insults caused by both HAV and HEV are important. It may be important to consider them in order to improve the prognosis of ACLF by developing a treatment for HAV infection.

## 3. HAV Infection and Metabolic Associated Fatty Liver Disease (MAFLD)

ACLF may occur among patients with chronic liver diseases or cirrhosis due to nonalcoholic fatty liver diseases (NAFLD), including NASH and alcoholic liver diseases (ALD), in eastern and western countries [2,13]. NASH is the most rapidly increasing etiology for ACLF [18]. Agrawal et al. reported a nonobese 34-year-old man presenting ACLF with acute HAV infection superimposed on NASH without cirrhosis [17] (Table 1). Kahraman et al. also reported a human immunodeficiency virus (HIV)-positive case presenting ACLF with acute HAV infection superimposed on cirrhosis due to NASH [19]. NASH is also observed among people less than 40 years old, and acute-on-chronic liver diseases may have an atypical course among these patients [20].

Fatty liver diseases associated with metabolic dysfunction are common and have a heterogeneous genetic predisposition, metabolic syndrome, and environmental factors [23]. Recently, experts suggested “MAFLD” should replace NAFLD/NASH [23]. The diagnosis of MAFLD is based on the detection of liver steatosis in the presence of overweight or obesity, diabetes mellitus, and/or clinical evidence of metabolic abnormalities, such as hypertension, dyslipidemia, and hyperglycemia.

A Japanese nationwide survey of ALF and late-onset hepatic failure (LOHF) caused by HAV infection suggested that diabetic mellitus was more common among deceased patients than among rescued patients (29% vs. 8%; *p* < 0.05), excluding patients with liver transplantations, and that diabetic mellitus was independently associated with the outcome [24]. Patients with diabetes are at risk of developing severe hepatitis [25].

We observed that HAV HA11-1299 genotype IIIA strain replication is enhanced by the accumulation of lipids or high-concentration glucose in the human hepatoma cell line, Huh7 [26,27,28]. Hyperglycemia or the accumulation of lipids induces an endoplasmic reticulum (ER) stress response in human hepatocytes. HAV replicates in the ER of human hepatocytes and induces an ER stress response. The ER stress response is mediated by the sensor molecules, inositol-requiring enzyme 1α (IRE1α), PKR-like ER kinase (PERK), and activating transcription factor 6 (ATF6), which are usually associated with molecular chaperone glucose-regulated protein 78 (GRP78) [27]. GRP78 is a negative regulator of ER stress response. We also observed that the overexpression of GRP78 could inhibit HAV replication, while the knockdown or knockout of GRP78 enhanced HAV replication [26,28]. In sum, GRP78 is an antiviral protein against HAV replication [28].

## 4. HAV Infection and Alcoholic Liver Diseases (ALD)

There are several factors affecting the severity of HAV infection and the rates of fulminant hepatic failure [29]. These important factors include older age, concomitant virus infection, chronic liver disease, sexual orientation, intravenous drug use, and alcohol abuse [6,29]. Feller et al. reported that 12 patients developed hepatic encephalopathy, ascites, or both, among 20 patients with alcoholic cirrhosis and a superimposed episode of acute viral hepatitis [30]. HAV infection was excluded in only three of these patients [30].

Lefilliatre et al. reported that three patients with fulminant hepatitis A had preexisting liver diseases, and one of the three had biopsy-proven alcoholic cirrhosis [21] (Table 1). Spada et al. reported that two individuals were HCV-coinfected alcohol abusers, had underlying liver cirrhosis, and died of acute liver failure due to HAV infection [22] (Table 1).

While the direct effects of alcohol on HAV replication is unknown, excess alcohol intake (binge drinking) could induce hepatic fibrosis. As only alcohol intake is responsible for worsening ACLF with alcoholic chronic liver diseases and alcoholic cirrhosis [31], HAV may have an additive responsibility for worsening ACLF with ALD.

## 5. Coinfection of HAV with HIV

In Japan, where no universal vaccination programs against HAV infection exist, 10–20% of those with HIV infection tested positive for immunoglobulin G (IgG) anti-hepatitis A (HA) antibodies [32,33,34,35]. This prevalence is similar to that of IgG anti-HA in those without HIV infection [36,37], although a higher prevalence area can also be observed in Japan [38]. In general, individuals of high-risk groups, such as healthcare workers, sewage workers, and drug addicts, have ~60% of IgG anti-HA worldwide [39,40]. The seroprevalence of IgG anti-HA is relatively higher in people living with HIV worldwide [41,42,43].

HIV infection has also been reported as a cause of liver damage in patients infected with HIV [40]. Thus, it is as important to consider patients with HIV infection as those with chronic liver disease. Not only chronic viral hepatitis B or C but also drug-induced liver injury induced by the antiretroviral drugs, NAFLD and ALD, has also been observed in people with HIV [40].

HAV infection in patients with chronic liver diseases and coinfected with HIV are shown in Table 2 [21,22,44,45]. Prolonged HAV infection was also reported in an HIV-seropositive patient [44]. It was reported that the recovery of immunity through recently developed anti-HIV therapies may lead to more severe hepatocellular damage in patients with HAV infection [45].

HAV infects humans through fecal–oral routes, when HAV-contaminated water and food are consumed. Among men who have sex with men (MSM), HAV is sexually transmitted [46], and HAV outbreaks have been observed [47,48,49,50,51,52,53,54,55]. It is noteworthy that acute hepatitis A among MSM is one of the male-dominant diseases, although, in general, no gender difference exists in patients with an HAV infection caused by HAV-contaminated water and food. While HAV may cause severe hepatitis in people living with HIV, two doses of an HAV vaccine are more effective for them to achieve a sustained HAV seroresponse than a single dose of an HAV vaccine [56].

## 6. Coinfection of HAV with HBV

Several cases of ACLF with HAV as an acute insult and chronic hepatitis or cirrhosis due to HBV, as well as cases with a superinfection of HAV in patients with HBV, have been reported (Table 3) [9,21,57,58,59,60,61,62]. A superinfection of HBsAg carriers with HAV seems not to cause more severe conditions [57]. Patients with HBV plus HAV infection had a less advanced baseline liver disease and a better prognosis than those with HBV plus hepatitis E virus infection [60].

Vento et al. reported that, among 10 patients with an acquired HAV superinfection and chronic HBV infection, one (10%), who had cirrhosis, had marked cholestasis [9]. Pramoolsinsap et al. evaluated acute superinfection with HAV in 20 HBV asymptomatic carriers and fulminant hepatitis or submassive hepatitis in 11 (55%) of 20 HBsAg carriers [63]. A superinfection of HAV in patients with HBV occasionally leads to critical conditions in HBV carriers with or without cirrhosis, although patients with advanced fibrosis or cirrhosis are more susceptible to severe conditions [9,21,57,58,59,60,61,62,63]. 

A total of 310,746 cases with acute hepatitis A were observed during the Shanghai hepatitis A epidemic [58]. A total of 47 fatal cases (0.015%) were reported. Fatality rates were 0.05% (15/27,346) and 0.009% (25/283,400) in patients with or without HBV infection, respectively. It is worth noting that there were 5.6-fold greater fatality rates in patients with HBV infection than in those without [58]. Cooksley et al. reported that patients infected with HBV who have raised ALT levels and high HBV levels have a higher risk of liver failure following HAV superinfection [58]. HAV vaccination seems to be effective in preventing liver failure associated with HAV in patients with or without HBV infection [64,65,66,67]. However, HAV vaccination may not be necessary in the case of countries in which HAV is endemic, such as India [68,69].

It has been reported that the transient suppression of HBV replication and the disappearance of HBV DNA with the seroconversion of HBeAg were observed in several cases of double infections with HAV and HBV carries [57]. Beisel et al. also reported that an HBsAg carrier case with HAV superinfection presented the seroconversion of HBsAg, suggesting that unspecific immunological responses to HAV could lead to a functional cure of HBV [62]. It was reported that the sharp peak in interferon-gamma production induced by a superinfection of HAV may lead to the suppression of HBV replication in patients with chronic hepatitis B [70]. This peak in interferon-gamma production occurred just before the rise in serum transaminase activity, resulting in a decrease in HBV DNA and HBeAg.

Berthillon et al. infected the human hepatoma cell line, PLC/PRF/5 [71], which integrates HBV DNA and produces HBsAg, with the HAV CF53 strain [72]. The inhibition of HBsAg production in PLC/PRF/5 cells infected with HAV was observed, compared with those without HAV infection, demonstrating that HAV interferes with the expression of HBsAg from hepatocytes harboring integrated HBV DNA sequences [71]. We also infected HepG2.2.15, which produces HBV virion or HepG2 cells, with the HAV HA11-1299 strain. We demonstrated that the HAV replication is similar between HepG2.2.15 and HepG2, 96 h after HAV infection. However, HBV replication is inhibited in HAV-infected HepG2.2.15, compared to HepG2.2.15 without HAV infection [73]. 

We also observed that the replication of both HAV and HBV is suppressed in human hepatocyte PXB cells superinfected with HAV and HBV, compared to those mono-infected with HAV or HBV [73]. Thus, HAV infection seems to inhibit HBV replication. Further studies are required to support this point, although it indicates that the existence of cirrhosis or advanced liver fibrosis should cause severe hepatitis in the superinfection of HAV in patients with HBV.

## 7. Coinfection of HAV with HCV

In general, HCV is a rare cause of fulminant hepatitis or acute liver failure [74,75]. We did not identify any cases of fulminant hepatitis with HCV RNA in 82 cases of fulminant hepatitis and late-onset hepatic failure from 1986 to 2001, which were examined at Chiba University School of Medicine, Japan [74]. There were several reports that HAV infection in patients with chronic hepatitis C is associated with increased mortality [9,21,22,76], although several contrary opinions exist [59,77] (Table 4).

Vento et al. reported that, among 17 patients with an acquired HAV superinfection with chronic hepatitis C, seven patients (41.2%) possessed fulminant hepatic failure, and six (85.7%) of those seven patients died [9]. It is interesting to note that antinuclear antibodies, anti-smooth-muscle antibodies, and/or anti-asialoglycoprotein receptor antibodies were detected in five of seven patients with fulminant hepatitis (71.4%) [9]. Moreover, six of these seven patients possessed chronic active hepatitis, and one patient recovered from fulminant hepatitis and was treated with methylprednisolone [9]. There are some reports indicating a higher fatality rate of HAV superinfection in patients with chronic HCV infection, not considering those with or without cirrhosis [21]. However, it is unclear whether the high fatality rates were due to severe underlying liver damage or not [21,22].

It was reported that the superinfection of HAV is associated with decreased HCV replication, which may lead to a clearance of HCV [77,78]. Esser-Nobis et al. found that Huh7-Lunet cells supported HAV and HCV replication with similar efficacy and limited interference with each other [79].

In fact, as several severe hepatitis A cases have been observed in patients with chronic HCV infection, clinicians should pay attention to HAV infection in HCV-infected individuals [80]. At present, although direct-acting antivirals against HCV can lead to a higher sustained virological response with less adverse events, no effective HCV vaccines are available. Thus, HAV vaccination should be considered for HCV-infected patients, especially those with cirrhosis or advanced fibrosis [81,82,83,84,85,86,87,88].

## 8. HAV and Other Chronic Liver Diseases

It was reported that a prospective study of 31 children in the age group of 1–16 years, who fulfilled the criteria for ACLF of the Asian Pacific Association for the Study of the Liver (APASL) 2008 consensus, found 13 ACLF cases of HAV as an acute insult and autoimmune hepatitis or Wilson disease as causes of chronic liver disease [15]. In children, acute-on-chronic liver diseases, HEV, and HAV are more frequently causes of acute insults and Wilson disease, while autoimmune liver disease and primary sclerosing cholangitis are more frequently causes of chronic liver disease [12]. It is possible that HAV infection, as an acute insult, could result in ACLF in patients with any chronic liver disease, especially cirrhosis. Careful attention should also be paid to HAV infection in adults and children who have certain chronic liver diseases.

## 9. Host Genetic Factors in HAV Infection

Acute insults in ACLF are different, depending on the country in which they are found [2]. In Asian countries, European countries, and the United States, hepatic, hepatic, and extrahepatic or infection (extrahepatic) causes, respectively, are representative acute insults in the definition of the APASL, EASL, and NACSELD ACLF guidelines [2,89,90,91]. Of course, not only a sanitary environment but also host genetic factors are different in these different regions. Among Mexican Americans, transforming growth factor beta 1 (TGFB1) rs1800469 (adjusted odds ratio (OR), 1.38; 95% confidence interval (CI), 1.14–1.68; P value adjusted for false discovery rate (FDR-P) = 0.017) and X-ray repair cross complementing 1 (XRCC1) rs1799782 (OR, 1.57; 95% CI, 1.27–1.94; FDR-P = 0.0007) were associated with an increased risk of HAV infection [92]. ATP-binding cassette subfamily B member 1 (ABCB1) rs1045642 (OR, 0.79; 95% CI, 0.71–0.89; FDR-P = 0.0007) was associated with a decreased risk [92]. Host genetic factors may also play an important role in determining the differential susceptibility to HAV infection [92,93,94].

## 10. Prevention of HAV Infection in Patients with Chronic Liver Diseases

### 10.1. HAV Vaccination

HAV vaccination may be important for patients with chronic liver diseases, especially those with cirrhosis [81,82,83,84,85,86,87,88]. While a universal vaccination program against HAV seems to be the most effective solution for the prevention of HAV infection, it may be difficult to carry out this program worldwide due to the high costs of HAV vaccine production and its low effectiveness in certain countries in which the infection is endemic [88,95]. HAV vaccination targeting certain populations may also be effective and important in this regard [96]. Antivirals against HAV infection may also be needed (Figure 1). The unknown causes for chronic injury constitute only 5–15% of cases of ACLF [2].

### 10.2. Japanese Rice-Koji Miso Extracts and Zinc Sulfates Could Inhibit HAV Replication with the Enhancement of GRP78 Expression

Japanese rice-koji miso extracts enhanced GRP78 expression and inhibited HAV HA11-1299 genotype IIIA strain replication in the human hepatocytes, Huh7 and PXB cells [97]. We investigated the effect of miso extracts on virus replication in HepG2.2.15 cells infected with the HAV HA11-1299 strain [73]. It is noteworthy that miso extracts have an inhibitory effect on HAV replication but no inhibitory effect on HBV replication. Japanese rice-koji miso extracts may have an inhibitory effect on HAV replication in patients superinfected with HAV and HBV.

The zinc homeostasis pathway was identified as a key pathway of the antiviral activity of Japanese rice-koji miso against HAV infection using transcriptome-sequencing analysis [98]. We also demonstrated that zinc sulfate has an inhibitory effect on HAV HA11-1299 replication in human hepatocytes with the enhancement of GRP78 expression [98]. As Japanese miso soup and zinc sulfate are traditional foods and drugs, respectively, they induce GRP78 expression and are useful and safe antiviral compounds against HAV, with fewer adverse events. Gut dysbiosis and increased permeability cause pathological bacterial translocation and endotoxemia, which play an important role in the development of ACLF [2]. HAV infects the liver by the gut-portal vein–liver axis through fecal–oral routes. The digestion and absorption of Japanese rice-koji miso extracts and zinc sulfate may be used through similar routes.

### 10.3. Candidates of Antivirals against HAV in Chronic Liver Diseases

The inhibitory effects of interferon-alpha, interferon-gamma, interferon-lambda, ribavirin, amantadine, sirtinol, and AZD1480 as host-targeting drugs and HAV 3C cysteine protease inhibitors, as well as small interfering RNAs against HAV, as antivirals that directly act on HAV replication, have been reported [11,46,99]. Interferon has antiviral potential against HAV [100,101], but it is difficult to use interferon in patients with ACLF, as interferon generally has cytotoxicity. Peginterferon-lambda has fewer side effects than peginterferon-alpha and may be useful in some patients with HAV infection. Amantadine is a broad-spectrum antiviral and has an inhibitory effect on HAV replication through the targeting of HAV internal entry site (IRES) activity [100,102,103]. The sirtuin inhibitor, sirtinol, also inhibits HAV replication by inhibiting HAV IRES activity [104]. Further studies on the mechanism of the sirtuin inhibitor and JAK pathways in HAV replication are needed [104,105]. In patients with chronic liver diseases or ACLF, these drugs should be improved, and more safe drugs are needed and should be explored. It has been reported that HCV receptor candidates, such as HAV cellular receptor 1 (HAVcr-1), integrin β1, and gangliosides, are the entry receptor candidates for HAV. Further studies in this vein are needed [106,107,108]. Gangliosides seem to function as endosome receptors for infection using both naked and quasi-enveloped HAV virions [108]. Blocking the cellular entry of HAV is also an attractive drug target for combating HAV infection.

### 10.4. HAV Infection Is Associated with the Activation of the Host Immune System and Severe Systemic Inflammation

Acute hepatitis A usually exhibits more severe inflammation, such as a higher fever and higher C-reactive protein levels, compared to acute hepatitis due to other hepatitis viruses [109,110,111]. Some cases of acute HAV infection present acute renal failure [112,113,114]. These results suggest that HAV infection activates human immune systems and induces cytokines [115,116,117,118,119]. Innate immunity also seems to be involved in the pathogenesis of hepatitis A [120,121]. Hypergammaglobulinemia and a high occurrence of autoantibodies are observed in HAV infection [122,123]. This may support the immunological basis of its pathogenesis. Moreover, the higher gammaglobulinemia in fulminant HAV suggests the existence of a more aggressive immunological reaction in severe hepatitis A [123]. 

Severe systemic inflammation can affect the functions of somatic cells in tissue and modify the clinical manifestation of cirrhosis and ACLF [124,125]. Patients with acute liver failure or ACLF are susceptible to infection, and early transplant-free survival is poor [126,127,128,129]. In liver transplantation for patients with ACLF, the role of the timing, bridging, and management of liver transplantation is important [130,131]. 

### 10.5. Recent Outbreak of HAV Infection in MSM

It has recently been reported that HAV susceptibility parallels the high COVID-19 mortality [132]. The 2019 coronavirus disease (COVID-19) has been observed in Japan, where the HAV susceptibility of the general population is high [34,35]. An HAV vaccination program is urgently required for individuals with or without HIV infection in this area. HAV infection is an imported infection, like novel severe acute respiratory syndrome coronavirus 2 (SARS-CoV-2) infection [133]. In the era of COVID-19, attention should also be paid to dual infection with HAV and SARS-CoV-2. 

An outbreak of HAV infection in MSM has been observed worldwide. An outbreak of acute HAV infection among HIV-coinfected MSM in Taiwan was observed from June 2015 to September 2017 [50,134,135]. Between July 2016 and February 2017, 48 male cases of HAV infection were found in the Netherlands [48]. A total of 17 of them were MSM. This strain is identical to a strain causing a large outbreak among MSM in Taiwan [48]. In the United States, HAV infections also increased among MSM from 2016 to 2018 [54,136,137]. Since 2017, HAV infection has increased among MSM in Japan [34,37,52]. RIVM-HAV16-090-like hepatitis A virus strains, which were >99.6% identical to the 66 reported strains isolated from Taiwan and European countries from 2015 to 2017, were also recovered from Japanese MSM [52]. A recent outbreak of HAV infection was also reported in various countries, such as Brazil, Spain, and Italy [53,138,139,140].

## 11. Possible Molecular Mechanism of the Development of ACLF in Patients with HAV Infection

The molecular mechanism of the development of ACLF in patients with HAV infection is not fully understood. The possible mechanisms of the development of liver failure in the presence of coinfection with HCV and HAV are as follows. HAV is a virus that is generally sensitive to interferon [100,101,102]. In comparison with HCV, HAV induces a limited production of type I interferon when HAV infects chimpanzees [141]. Compared with HBV and HCV, HAV weakly induced the activation of NF-κB signaling pathways in human hepatocytes [142,143]. While HAV VP3 activates cell growth signaling [143], HAV VP1/2A reduces cell viabilities in HCV sub-genomic replicon cells [144]. 

HAV is usually a non-cytopathic virus, and HAV inhibits double-stranded (dsRNA)-induced interferon-beta gene expression by influencing the interferon-beta enhanceosome, as well as dsRNA-induced apoptosis [145]. Compared with HBV and HCV, HAV could evade mitochondrial antiviral signaling protein (MAVS)-mediated type I interferon responses [146]. HAV 3ABC is capable of MAVS cleavage, like HCV NS3/4A, which cleaves MAVS and disrupts interferon signaling [147]. HAV 3C inhibits HAV IRES-dependent translation and cleaves the polypyrimidine tract-binding protein [148]. HCV induces interferon-beta signaling pathways in human hepatocytes [149]. Controlling the effects of interferon signaling may determine the prognosis of patients coinfected with HCV and HAV (Figure 2). 

HBV is a stealth virus which efficiently infects humans without alerting the innate immune system, although HCV strongly induces but cunningly evades the innate immune response [150]. The high glucose and fat deposition of hepatocytes seem to induce a chaperon-mediated autophagy (CMA) [151]. CMA targets interferon-alpha receptor chain-1 for degradation, dampens hepatic innate immunity, and disrupts interferon signaling pathways [151]. CMA is also observed in patients with ALD or MAFLD [152,153]. Altering interferon signaling may contribute to ALF-associated acute HAV infection. However, further studies are needed. Among HIV-positive patients with acute HAV infection, lower peaks in total bilirubin, AST, and ALT levels were observed in comparison with HIV-negative patients with acute HAV infection [154], suggesting that weaker immune responses occur in HIV-positive patients. These immune responses could enhance HAV replication and modify the pathogenesis in HIV-positive patients with acute hepatitis A [155]. 

## 12. Conclusions

We reviewed the literature concerning HAV infection in patients with chronic liver diseases. In patients with chronic liver diseases, HAV infection can occasionally lead to a critical condition, such as acute liver failure. There seems to be no etiological association between liver failure and HAV infection, but there is a significant correlation between the severity of liver disease and the degree to which the liver has already been damaged. While there are effective HAV vaccines currently in existence, antivirals against HAV should be further explored. The latter is urgent given that the lives of patients with HAV infection and a chronic liver disease of another etiology may be at immediate risk.

## Figures and Tables

**Figure 1 ijms-21-06384-f001:**
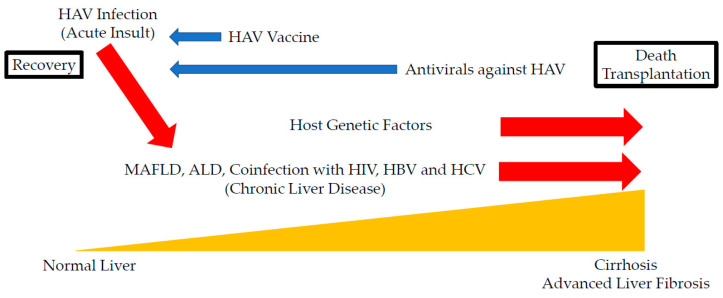
Effects of hepatitis A infection (HAV) on the prognosis of chronic liver disease. Possible acceleration and inhibition of the disease progression of hepatitis A are indicated by red and blue arrows, respectively. MAFLD, metabolic associated fatty liver disease; ALD, alcoholic liver disease; HIV, human immunodeficiency virus; HBV, hepatitis B virus; HCV, hepatitis C virus.

**Figure 2 ijms-21-06384-f002:**
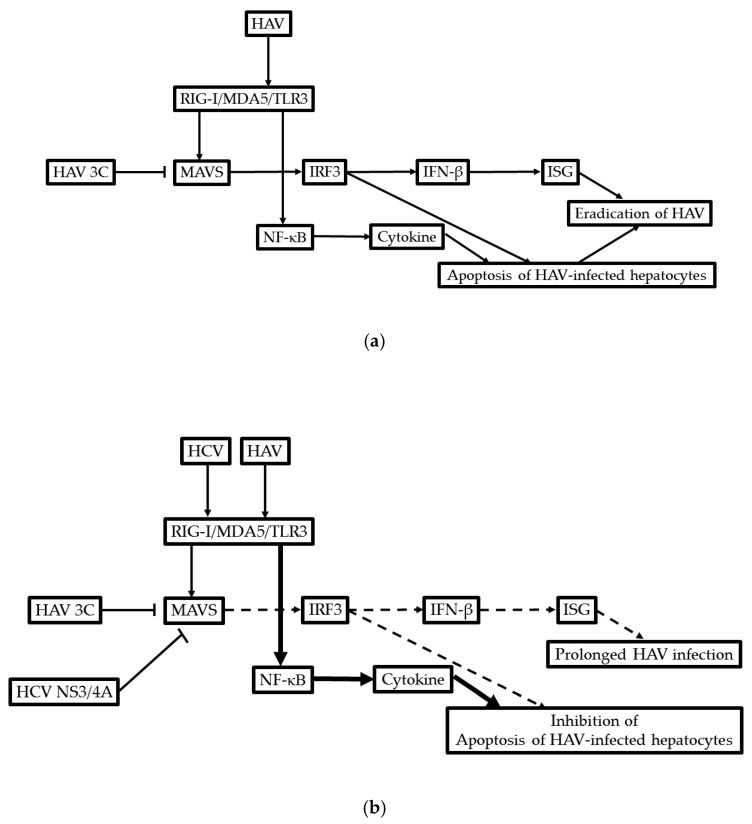
Possible molecular mechanism of the development of acute-on-chronic liver failure (ACLF) in patients coinfected with hepatitis A virus (HAV) and HCV. (**a**) Only HAV infection; (**b**) coinfection HAV and HCV. RIG-I, retinoic acid-inducible gene-I; MDA-5, melanoma differentiation associated gene 5; TLR3, toll-like receptor 3; MAVS, mitochondrial antiviral signaling protein; IRF3, interferon regulatory factor 3; IFN, interferon; ISG, interferon-stimulated gene; NF-κB, nuclear factor kappa B subunit 1.

**Table 1 ijms-21-06384-t001:** Acute-on-chronic liver failure with hepatitis A virus (HAV) infection in patients with nonalcoholic steatohepatitis (NASH) or chronic alcoholic liver diseases (ALD).

Authors (Year) [References]	N	Acute Insults	Underlying CLD	Prognosis
Agrawal S, et al. (2018) [17]	1	HAV	NASH	Recovered
Kahraman A, et al. (2006) [19]	1	HAV	NASH and HIV	Died
Lefillatre P, et al. (2000) [21]	1	HAV	ALD	Died
Spada E, et al. (2005) [22]	2	HAV	ALD and HCV	Died

CLD, chronic liver diseases; HIV, human immunodeficiency virus; ALD, alcoholic liver disease; HCV, hepatitis C virus.

**Table 2 ijms-21-06384-t002:** Coinfection with hepatitis A virus (HAV) and human immunodeficiency virus (HIV).

Authors (Year) [References]	N	Acute Insults	Underlying CLD	Prognosis
Lefillatre P, et al. (2000) [21]	1	HAV	HBV, HCV, and HIV	Died
Spada E, et al. (2005) [22]	1	HAV	HCV and HIV	Died
Costa-Mattioli et al. (2002) [44]	1	HAV	HIV	Alive; HAV RNA detected in 256 days
Maki Y, et al. (2020) [45]	1	HAV	HIV	Died

CLD, chronic liver diseases; HBV, hepatitis B virus; HCV, hepatitis C virus.

**Table 3 ijms-21-06384-t003:** Acute-on-chronic liver failure and/or superinfection of hepatitis virus (HAV) in patients with hepatitis B virus (HBV).

Authors (Year) [References]	N	Acute Insults	Underlying CLD	Prognosis
Tassopoulos N, et al. (1985) [57]	10	HAV	HBV	Recovered
Vento S, et al. (1998) [9]	10	HAV	HBV	Recovered (marked cholestasis, 1)
Lefillatre P, et al. (2000) [21]	1	HAV	HBV	Died
Cooksley WGE, et al. (2000) [58]	27,346	HAV	HBV	Died, 15 (0.05%)
Sagnelli E, et al. (2006) [59]	13	HAV	HBV	Recovered (severe hepatitis, 1)
Zhang X, et al. (2010) [60]	52	HAV	HBV	Died, 1 (1.9%) [Hepatic failure, 6 (11.5%)]
Fu J, et al. (2016) [61]	35	HAV	HBV	Recovered
Beisei C, et al. (2020) [62]	1	HAV	HBV	Recovered (seroconversion of HBeAg to anti-HBe)
Lefillatre P, et al. (2000) [21]	1	HAV	HBV, HCV, and HIV	Died

CLD, chronic liver diseases; HCV, hepatitis C virus; HIV, human immunodeficiency virus; HBeAg, hepatitis B virus e antigen.

**Table 4 ijms-21-06384-t004:** Acute-on-chronic liver failure (ACLF) and/or the superinfection of hepatitis virus (HAV) in patients with hepatitis C virus (HCV).

Authors (Year) [References]	N	Acute Insults	Underlying CLD	Prognosis
Vento S, et al. (1998) [9]	17	HAV	HCV	Recovered, 10; fulminant hepatitis, 7
Sagnelli E, et al. (2006) [59]	8	HAV	HCV	Recovered
Deterding K, et al. (2006) [77]	17	HAV	HCV	Fulminant hepatitis, 0
Spada E, et al. (2005) [22]	1	HAV	HCV and ALD	Died
Spada E, et al. (2005) [22]	1	HAV	HCV and HIV	Died
Lefillatre P, et al. (2000) [21]	1	HAV	HBV plus HCV and HIV	Died

CLD, chronic liver diseases; HBV, hepatitis B virus; ALD, alcoholic liver disease; HIV, human immunodeficiency virus.

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
