# Peer review of "Co-Occurrence of Hepatitis A Infection and Chronic Liver Disease"

_ijms, 2020, doi:10.3390/ijms21176384_

Round 1

Reviewer 1 Report

This manuscript has described the pathological conditions of Hepatitis A infection related to various liver diseases. The authors have discussed numerous previous literature to provide deep and broad research information about Hepatitis A. 

However, the manuscript has described a lot of clinical cases about hepatitis A with other liver diseases. Thus, the reviewer should describe 1) the molecular actions of the viral infection of hepatitis A to provoke hepatic pathology, 2) detailed molecular pathways of how hepatitis A aggravates hepatic pathology with various viral infections, including HIV and HBV and HCV.

While the title of the manuscript is "Hepatitis A infection in chronic liver diseases", the portion of manuscript describing HAV with chronic liver diseases (Alcoholic liver disease, NASH or NAFLD) is too small. Thus, the authors should provide more scientific information and inputs on the pathology of HAV in chronic liver diseases, including alcoholic steatohepatitis, NASH and/or NAFLD. 

Finally, the authors had better to provide graphical models that will help the general audience to understand the concept of HAV in chronic liver diseases much easier.

Author Response

To Reviewer #1: Thank you for your valuable comments and criticisms.

Response to your comment: “However, the manuscript has described a lot of clinical cases about hepatitis A with other liver diseases. Thus, the reviewer should describe 1) the molecular actions of the viral infection of hepatitis A to provoke hepatic pathology, 2) detailed molecular pathways of how hepatitis A aggravates hepatic pathology with various viral infections, including HIV and HBV and HCV.”

Thank you for your comments and we agree with you. Accordingly, we added a new Figure 2 and revised our manuscript as follows.

In page 9, line 328 – page 10, line 353,

  1. Possible molecular mechanism of the development of ACLF in patients with HAV infection

                 The molecular mechanism of the development of ACLF in patients with HAV infection is not fully understood. The possible mechanisms of the development of liver failure in the presence of a coinfection with HCV and HAV are as follows. HAV is a virus that is generally sensitive to interferon [100-102]. In comparison with HCV, HAV induces a limited production of type I interferon when HAV infects chimpanzees [141]. Compared with HBV and HCV, HAV weakly induced the activation of NF-κB signaling pathways in human hepatocytes [142, 143]. While HAV VP3 activates cell growth signaling [143], HAV VP1/2A reduces cell viabilities in HCV sub-genomic replicon cells [144].

HAV is usually a non-cytopathic virus, and HAV inhibits double-stranded (dsRNA)-induced interferon-beta gene expression by influencing the interferon-beta enhanceosome, as well as dsRNA-induced apoptosis [145]. Compared with HBV and HCV, HAV could evade mitochondrial antiviral signaling protein (MAVS)-mediated type I interferon responses [146]. HAV 3ABC is capable of MAVS cleavage, like HCV NS3/4A, which cleaves MAVS and disrupts interferon signaling [147]. HAV 3C inhibits HAV IRES-dependent translation and cleaves the polypyrimidine tract-binding protein [148]. HCV induces interferon-beta signaling pathways in human hepatocytes [149]. Controlling the effects of interferon signaling may determine the prognosis of patients coinfected with HCV and HAV (Figure 2).

HBV is a stealth virus, which efficiently infects humans, without alerting the innate immune system, although HCV strongly induces but cunningly evades the innate immune response [150]. The high glucose and fat deposition of hepatocytes seem to induce a chaperon-mediated autophagy (CMA) [151]. CMA targets interferon-alpha receptor chain-1 for degradation, dampens hepatic innate immunity, and disrupt interferon signaling pathways [151]. CMA is also observed in patients with ALD or MAFLD [152,153]. Altering interferon signaling may contribute to ALF-associated acute HAV infection. However, further studies are needed. Among HIV-positive patients with acute HAV infection, lower peaks of total bilirubin, AST, and ALT levels were observed and compared with HIV-negative patients with acute HAV infection [154], suggesting that weaker immune responses occur in HIV-positive patients. These immune responses could enhance HAV replication and modify the pathogenesis in HIV-positive patients with acute hepatitis A [155].

Response to your comment: “While the title of the manuscript is "Hepatitis A infection in chronic liver diseases", the portion of manuscript describing HAV with chronic liver diseases (Alcoholic liver disease, NASH or NAFLD) is too small. Thus, the authors should provide more scientific information and inputs on the pathology of HAV in chronic liver diseases, including alcoholic steatohepatitis, NASH and/or NAFLD.”

Thank you for your comments and we agree with you. Accordingly, we made change of title to “Co-occurrence of Hepatitis A infection and chronic liver disease” and extensively revised the manuscript.

Response to your comment: “Finally, the authors had better to provide graphical models that will help the general audience to understand the concept of HAV in chronic liver diseases much easier.”

Thank you for your comments and we agree with you. Accordingly, we made a new Figure 2.

Reviewer 2 Report

Tatsuo Kanda et al. review in their manuscript co-occurrences of Hepatitis A infection and chronic liver disease. They argue that an effective treatment for Hepatitis A is required because when it occurs in conjunction with a chronic liver disease of another aetiology the patient's life might be at immediate risk. 

The authors list a lot of different cases of chronic liver disease and Hepatitis A infections but I fail to see a coherent message. The conclusion is similar to that patients already affected with a chronic liver disease are more likely to develop a severe liver failure scenario when affected by an Hepatitis A infection. 

I think the manuscript is laking a concise message beyond this relatively obvious information. It is difficult to read and should be restructured to make the different aetiologies more comparable (if this is possible). For instance in the sense: which aetiology of a chronic liver disease is particularly prone to develop into a liver failure when the patient is simultaneously affected by a Hepatitis A infection. Or: is there no correlation between aetiology developing a liver failure when affected by the Hepatitis A virus ?  - is the correlation more with the level the liver is already damaged ?

Author Response

To Reviewer #2: Thank you for your valuable comments and criticisms.

Response to your comment: “The authors list a lot of different cases of chronic liver disease and Hepatitis A infections but I fail to see a coherent message. The conclusion is similar to that patients already affected with a chronic liver disease are more likely to develop a severe liver failure scenario when affected by an Hepatitis A infection.”

Thank you for your comments and we agree with you. Accordingly, we revised our manuscript as follows.

In page 1, lines 19 – 25,

…these conditions and mechanisms. There may be no etiological correlation between liver failure and HAV infection, but there is an association between the level of chronic liver damage and the severity of acute-on-chronic liver disease. While the application of an HAV vaccination is important for preventing HAV infection, the development of antivirals against HAV may be important for preventing the development of ACLF with HAV infection as an acute insult. The latter is all the more urgent, given that the lives of patients with HAV infection and a chronic liver disease of another etiology may be at immediate risk.

In lines 363 – 370,

  1. Conclusions

We reviewed the literature concerning HAV infection in patients with chronic liver diseases. In patients with chronic liver diseases, HAV infection can occasionally lead to a critical condition, such as acute liver failure. There seems no etiological association between liver failure and HAV infection, but there is a significant correlation between the severity of liver disease and the degree to which the liver has already been damaged. While there are effective HAV vaccine currently in existence, antivirals against HAV should be further explored. The latter is urgent, given that the lives of patients with HAV infection and a chronic liver disease of another etiology may be at immediate risk.

Response to your comment: “I think the manuscript is laking a concise message beyond this relatively obvious information. It is difficult to read and should be restructured to make the different aetiologies more comparable (if this is possible). For instance in the sense: which aetiology of a chronic liver disease is particularly prone to develop into a liver failure when the patient is simultaneously affected by a Hepatitis A infection. Or: is there no correlation between aetiology developing a liver failure when affected by the Hepatitis A virus ?  - is the correlation more with the level the liver is already damaged ?”

Thank you for your comments and we agree with you. Accordingly, we revised our manuscript as follows.

In page 1, lines 19 – 25,

…these conditions and mechanisms. There may be no etiological correlation between liver failure and HAV infection, but there is an association between the level of chronic liver damage and the severity of acute-on-chronic liver disease. While the application of an HAV vaccination is important for preventing HAV infection, the development of antivirals against HAV may be important for preventing the development of ACLF with HAV infection as an acute insult. The latter is all the more urgent, given that the lives of patients with HAV infection and a chronic liver disease of another etiology may be at immediate risk.

In lines 363 – 370,

  1. Conclusions

We reviewed the literature concerning HAV infection in patients with chronic liver diseases. In patients with chronic liver diseases, HAV infection can occasionally lead to a critical condition, such as acute liver failure. There seems no etiological association between liver failure and HAV infection, but there is a significant correlation between the severity of liver disease and the degree to which the liver has already been damaged. While there are effective HAV vaccine currently in existence, antivirals against HAV should be further explored. The latter is urgent, given that the lives of patients with HAV infection and a chronic liver disease of another etiology may be at immediate risk.

Round 2

Reviewer 1 Report

The authors did not spare any efforts to revise the manuscript. This manuscript is now suitable for publication in IJMS.